# Viral Metagenomics Reveals a Putative Novel HPV Type in Anogenital Wart Tissues

**DOI:** 10.3390/pathogens11121452

**Published:** 2022-12-01

**Authors:** Huimin Hu, Yu Ling, Xuan Wang, Hao Wang, Niannan Zhu, Yumei Li, Hui Xu

**Affiliations:** 1Department of Dermatology, Affiliated Hospital of Jiangsu University, Zhenjiang 212001, China; 2Department of Dermatology, The Affiliated Huai’an Hospital of Xuzhou Medical University and The Second People’s Hospital of Huai’an, Huaian 223002, China; 3Department of Clinical Laboratory, Women’s Hospital of Nanjing Medical University, Nanjing 210004, China

**Keywords:** anogenital warts, putative novel human papillomavirus, *Gammapapillomavirus*, viral metagenomics

## Abstract

Viral metagenomics is widely employed to identify novel viruses in biological samples. Recently, although numerous novel human papillomavirus (HPV) types have been identified in clinical samples including anogenital warts (AGWs), many novel HPV sequences remain to be discovered. In this study, a putative novel HPV type designated as HPV-JDFY01 was discovered from library GW05 with 63 sequence reads by the viral metagenomic technique. Its complete genomic sequence was determined by PCR to bridge the gaps between contigs combining Sanger sequencing. The complete genome of HPV-JDFY01 is a 7186 bp encoding 7 open reading frames (ORFs) (E6, E7, E1, E2, E4, L2 and L1) and contains a 487 bp long control region (LCR) between L1 and E6. Sequence and phylogeny analysis indicated that HPV-JDFY01 shared the highest sequence identity of 74.2% with HPV-mSK_244 (MH777383) and well clustered into the genus *Gammapapillomavirus*. It has the classical genomic organization of *Gammapapillomaviruses*. Epidemiological investigation showed that one out of the 413 AGW tissue samples was positive for HPV-JDFY01. Further research with large size and different type of samples should be performed to elucidate the epidemiologic status of HPV-JDFY01.

## 1. Introduction

HPVs are present as small, nonenveloped and circular double-stranded DNA viruses with an icosahedral capsid, and they consist of an average genome length of 8000 bp [1,2]. The whole genome can be broadly grouped into early regions, late regions and a long control region according to their different functions [3,4,5]. The outer capsid of HPV is comprised predominantly of seventy-two major capsid protein L1, which plays an essential role in the attachment during the viral entry and the assembling of virions [6,7]. L1 protein has also been applied for the classification or defining of a novel HPV type if the variability proportion is over 10% [3].

The International Human Papillomaviruses Reference Center has confirmed 223 HPV types with complete genome as of 5/April/2022 <https://www.hpvcenter.se/>. These HPVs are only distributed over five evolutionary genera, including *Alphapapillomavirus*, *Betapapillomavirus*, *Gammapapillomavirus*, *Mupapillomavirus* and *Nupapillomavirus*.

*Gammapapillomavirus* accounts for over 50% of total HPVs and owns 83.3% of the novel types from 2014, which demands significant attention [8]. Little is known about its genetic characteristics, but it has been reported that all of the Gamma-6 lack the E6 protein to promote carcinogenesis [9]. In addition, patients with warts, hypogammaglobulinemia, infections or myelokathexis (WHIM) syndrome seem to be inclined to be infected by *Gammapapillomavirus* [10].

In the past four years, only four novel HPV types had been identified, including HPV226 (Gamma-6), HPV227 (Beta-2), HPV228 (Gamma-27) and HPV229 (Gamma-7) (https://www.hpvcenter.se/), three of which belonged to *Gammapapillomavirus*. Latsuzbaia et al. [11] discovered that the strain HPV226 contained one TATA box and three palindromic E2-binding sites in LCR, one zinc-binding domain in E7 and one ATP-binding site in E1. Other special structural features, such as polyadenylation sites and LxCxE motif were identified in HPV227 [12]. Both HPV226 and HPV227 were detected by next-generation sequencing (NGS) after rolling circular amplification [11,12].

Viral Metagenomics is based on NGS and can detect all the viral nucleic acid directly in biological samples so as to effectively find possible new viruses. In our study, using viral metagenomic method, a putative novel HPV (HPV-JDFY01) was discovered from AGWs tissue samples and the complete genome was acquired by combining PCR and Sanger sequencing. Sequence and phylogeny analysis were further performed to fully characterized this putative novel type of HPV.

## 2. Materials and Methods

### 2.1. Sample Collection and Preparation

A total of 110 tissue samples of wart from AGW patients, including 34 females and 76 males, were collected from September 2016 to August 2017 from the Department of Dermatology and Venerology at the Affiliated Hospital of Jiangsu University. The mean age of AGW patients was 42.5 years and the mean disease duration was 3.26 months. Each sample was transferred into 1.5 mL test tube with 1 mL dulbecco’s phosphate-buffered saline (DPBS) and then added into several 5 mm sterile stainless-steel beads to fully grind tissue for 5 min. After freeze and thaw 3 times on dry ice for 3 min each time, the tissue supernatants were obtained after centrifuging for 10 min at 12,000× *g* under the conditions of 4 °C.

### 2.2. Viral Metagenomic and Bioinformatics Analysis

The 110 supernatants of wart tissues were randomly pooled into 11 sample pools, each containing 10 samples. These sample pools were filtered through a 0.45 μm filter (Merck Millipore Ltd., Tullagreen, Carrigtwohill, Darmstadt, Germany, Co. Cork, IRL) to remove cellular particles in the sample [13,14]. The filtrates were digested by DNase and RNase at 37 °C for 60 min to remove non-viral encapsidated nucleic acids [15,16]. Then, a QIAamp Viral RNA Mini Kit (QIAGEN GmbH, QIAGEN Strasse 1, 40724 Hilden, Germany) was utilized to extract the remaining nucleic acids in the samples [17]. Reverse transcription (RT) and Klenow reactions were conducted to synthesize double-stranded DNA, which was prepared for constructing the 11 cDNA libraries by Nextera XT DNA Sample Preparation Kit (Illumina, Inc., San Diego, CA 92122 USA) [18,19]. Libraries were subjected to NGS on the MiSeq Illumina sequencing platform. The generated NGS data was processed by in-house analysis pipeline. Briefly, the bioinformatics steps were performed on a cluster of 32 nodes running Linux, and taking the Phred Score Q30 as the threshold trimmed the low sequencing quality tails [20]. Adaptors and primer sequences were trimmed by VecScreen in the default parameters [21]. These cleaned reads from sequencing were de novo assembled into contigs within each barcode group by Ensemble assembler [22]. The assembled contigs, along with unassembled reads, were matched against viral proteome database by BLASTx with an E-value cutoff of <10^−5^, where the viral proteome databased included NCBI virus reference proteome together with viral proteins sequences from NCBI nr fasta file.

### 2.3. Acquiring of Complete Genome and PCR Reactions

Finding six contigs ranging from 123 bp to 645 bp in length in GW05 pool, which showed low sequence similarity to known HPVs, 2 sets of specific PCR primers derived from a contig belonging to L1 gene of HPV were, respectively designed to confirm the existence of the putative novel HPV sequence in samples from GW05 pool. PCR primer sequences are displayed in Table 1. The screening conditions of PCR were as follows: 95 °C for 5 min, 31 cycles of 95 °C for 30 s, 48 °C (for the first round) or 52 °C (for the second round) for 30 s and 72 °C for 40 s and a final extension at 72 °C for 5 min. The circular complete genome of HPV-JDFY01 was acquired by PCR primers bridging the gaps between contigs and the PCR products were cloned and subjected to Sanger sequencing. In order to assess the prevalence of HPV-JDFY01 in AGW samples, an epidemiological investigation was performed by PCR screening method described above for a total of 413 AGW specimens collected from August 2017 to August 2020.

### 2.4. Phylogenetic Analysis

Phylogenetic analysis was performed based on amino acid sequences of predicted ORFs to analyze the genetic relationships between HPV-JDFY01 and the other related HPV types which included the representative types of *Alphapapillomavirus*, *Betapapillomavirus* and *Gammapapillomavirus*. Multiple alignments were conducted using ClustalW in MEGA7 with the default settings [23]. The aligned file was converted to Nexus (.nex) formmat by Geneious 11.1.2 and subsequently modified by Notapade++ [24]. The mean standard deviation of split frequencies was controlled as low as possible when generating phylogenetic trees by Mrbayes 3.2.7 with Bayesian inference (BI) [25].

## 3. Results

The NGS results indicated that the 11 libraries generated a total of 6,509,970 reads, 726,777 (11.16%) of which showed significant sequence similarity to HPV. The library GW02 generated the largest number of total reads (1,300,146) while the library GW09 contained the most abundant HPV-related sequence reads (227,374). Among the HPV sequence reads in these libraries, Alpha-10 (26,708) ranking first was followed by Alpha-8 (20,929), Alpha-14 (1622) and Alpha-9 (1066). HPV11 occupied the dominant part in Alpha-10 (95.41%) and HPV of the GW05 library (33.87%), respectively, in accordance with that of the literature reported previously [26,27]. HPV6, another common subtype of AGW, accounted for 0.42% in GW05 library. Except for unclassified papillomaviridae, the other HPV types identified in the 110 wart samples were presented in Appendix A. Given that six contigs ranging from 123 bp to 645 bp in length in the GW05 pool showed low sequence similarity to known HPVs, 10 individual samples in this pool were screened by the nested PCR. The result indicated only one sample was positive. PCR combining Sanger sequencing was then used to bridge the sequence gaps and obtain the whole genome of the putative novel HPV eventually.

### 3.1. Complete Genomic Structure of HPV-JDFY01

A putative novel HPV type (HPV-JDFY01) was discovered from AGW, of which the complete circular genome is 7186 bp in length with a GC content of 36.6%. The details of the seven predicted ORFs are described as follows: Early proteins: E6 (nucleotide (nt) 1–420, 420 bp), E7 (nt 417–701, 285 bp), E1 (nt 688–2514, 1827 bp), E2 (nt 2447–3604, 1158 bp) and E4 (nt 3048–3374, 327 bp); Late proteins: L2 (nt 3606–5135, 1530 bp) and L1 (nt 5146–6699, 1554 bp); and a 487 bp long control region (LCR) is located at 6700 to 7186 of the genome between L1 and E6 (Figure 1). The putative E6 protein of HPV-JDFY01 contains two conserved zinc-finger domains, CxxC(x)29CxxC being located at nucleotide positions of 79–189 and 298–408, which are separated by 36 amino acids [28,29]. Three putative DZ binding domains (PTAL, nucleotide positions 16–27; LSLV, nucleotide positions 139–150 and ESLV, nucleotide positions 256–267) were also identified [30]. The putative E7, a downstream protein of E6, contains one zinc-finger domain (nucleotide positions 558–668), but no binding domain (LxCxE) for the conserved retinoblastoma tumor suppressor protein (pRB) [31]. As the largest protein of HPV-JDFY01, the putative E1 has one conserved ATP-binding site of the ATP-dependent DNA helicase with a consensus sequence (GPPDTGKS, nucleotide positions 1966–1989) and one typical bipartite nuclear localization signal (NLS) (KRK, nucleotide positions 919–927; KRRL, nucleotide positions 1009–1020). A leucine-rich nuclear export signal (NES) (LSPRLEAVKI, nucleotide positions 961–990) situates between the two clusters of NLS [32]. The putative E4 protein, with the highest proline content (12.8%), lies fully inside the E2. One highly conserved furin cleavage motif (RAKR, nucleotide positions 3620–3631) has been found at the N-terminal part of the L2 protein. In addition, a putative polyadenilation site (AATAAA, nucleotide positions 3695–3700) and a L2 transmembrane domain-like aa sequence (IIYLGGLGIGTGKGTG, nucleotide positions 3749–3796) are also observed [33]. NLS-like signals, which are used to direct HPV virions to import into the nucleus, are identified both at the C-terminal part of L2 protein (nucleotide positions 5099–5119) and in L1 of HPV-JDFY01 (nucleotide positions 6622–6696). The 487-bp LCR, also known as upstream regulatory region (URR), contains one putative polyadenilation site (AATAAA, nucleotide positions 6754–6759) and three palindromic E2-binding sites (ACC-AGTTCT-GGT, nucleotide positions 6914–6925; ACC-GATAAC-GGT, nucleotide positions 6955–6966 and ACC-AAGTGT-GGT, nucleotide positions 7135–7146) [34]. A TATA box (TATAAA, nucleotide positions 7150–7155) is located at the 31 nt upstream of E6’s first start codon. Most of the above-mentioned genetic characteristics could be examined in all other Gamma-8 species members. In view of the highest sequence identity of HPV-JDFY01 was only 74.2% with HPV-mSK_244 (MH777383), it needs to be confirmed as one putative novel HPV type by the current demarcation criteria of International Committee on the Taxonomy of Viruses (ICTV) [1,3].

### 3.2. Phylogenetic Analysis of HPV-JDFY01

The amino acid sequence of HPV-JDFY01 L1 was aligned with 11 typical reference sequences from *Alphapapillomavirus*, 9 from *Betapapillomavirus*, 15 from *Gammapapillomavirus* and 13 from unclassified papillomavirus so as to generate the phylogenetic tree that clearly reflected the evolutionary relationships of these representative HPV types. From the results of the phylogenetic analysis, the HPV-JDFY01, together with HPV-mSK_244 (Figure 2, Appendix A) was clustered into the same clade of *Gammapapillomavirus*, a genus with the highest number of novel HPVs in recent years [8]. Both HPV-JDFY01 and HPV-mSK_244 were directly adjacent to the HPV112 which belongs to the Gamma-8 species. The sequence identity and phylogenetic analysis supported that HPV-JDFY01 was closely related to Gamma-8 types.

### 3.3. Epidemiological Analysis of HPV-JDFY01

Epidemiological investigation showed that only one out of the 413 AGW tissue samples was positive for HPV-JDFY01. The typically clinical and pathological manifestations were shown in Figure 3A,B.

### 3.4. GenBank Accession Number

The complete genomic sequence of HPV-JDFY01 was submitted to the GenBank database under accession number MW311489.

## 4. Discussion

Viral metagenomics, based on NGS, can detect nucleic acid of all viral pathogens from samples and have the potential to become crucial for clinical diagnosis in the near future [35,36,37,38]. We report a genome of HPV-JDFY01, a putative novel type in *Gammapapillomavirus*, identified by viral metagenomics from AGWs collected in Jiangsu Province, China.

Previous literatures considered that HPVs of the same species share ≥ 71% L1 similarity [1,3]. Phylogenetic analysis indicated that HPV-JDFY01 shared the highest sequence identity of 74.2% with HPV-mSK_244 based on the nucleotide sequence of L1. Thus, HPV-JDFY01 could be assumed as a putative novel HPV type and may have similar biological functions with HPV-mSK_244. HPV-mSK_244 was identified from 207 skin and nares swabs from DOCK8-deficient patients, who have recurrent cutaneous and systemic infections and can suffer from the predisposition and susceptibility to cancer [7]. However, HPV-JDFY01 was found in immunocompetent AGW patients. Additionally, in the phylogenetic analyses of L1 protein, HPV-JDFY01 and HPV-mSK_244 were clustered together with HPV112, which belongs to the Gamma-8 species. It is worth noting that HPV112, a member of the Gamma-8 species, was also cloned from a 25-year-old male with condyloma acuminata [39]. The other two types of Gamma-8, HPV164 and HPV168, were isolated from exfoliated skin cell samples of healthy individuals [40,41]. In summary, the above evidences indicate that HPV types from the Gamma-8 species may engage in much milder relationships with human beings. Additionally, the relative abundances of DNA viruses, mostly *Papillomaviridae* and *Polyomaviridae*, were higher on the skin than in the oral cavity and feces of DOCK8-deficient patients [7]. HPV-JDFY01 was isolated from tissue samples of wart while HPV112, HPV164 and HPV168 were all isolated from skin [39,40]. Therefore, it can be stated that the members of Gamma-8 tend to infect skin and may be distinguished into cutaneotropic HPV types.

Even though further studies are needed to assess the relationship between HPV-JDFY01 and Gamma-8, the genetic characteristics of HPV-JDFY01 were further analyzed by comparing with the previous literatures, which were related to other HPV members of Gamma-8. We found that most of specific characteristics of Gamma-8 could be identified in HPV-JDFY01, including zinc-finger domains, PDZ binding domains, ATP-binding sites, nuclear localization signals, nuclear export signals, NLS-like signals, furin cleavage motifs, polyadenilation sites, palindromic E2-binding sites and TATA boxes. In previous literature, it has been reported that the E7 protein of HPV112 contained a pRB protein binding domain, although it was not totally conserved [39]. However, pRB protein binding motif (LxCxE) have not been detected in E7 protein of HPV-JDFY01. This means lacking the LxCxE motif may have the effect of attenuating the association between E7 and pRB [42] and indicate transforming properties of HPV-JDFY01. A highly conserved NLS motif can promote nuclear localization and bind to the nuclear matrix [43], which was also not identified in E2 of HPV-JDFY01. Moreover, three internal PDZ-binding motifs (PTAL, LSLV and ESLV) were identified in the putative E6 protein. Nevertheless, these motifs may not possess biological functions [44]. Further structural analysis will be needed to better understand the relationship between HPV-JDFY01 and the other members from Gamma-8.

On the basis of the highly conserved, immunogenic and self-assembly natures in HPV L1 protein, virus-like particles (VLPs) of HPV have been designed and developed for the purpose of prophylactic HPV vaccines [45]. Further prediction analyses of advanced structure for novel HPV L1 protein are thus warranted.

## 5. Conclusions

In this study, a putative novel *Gammapapillomavirus* named HPV-JDFY01 was detected by a viral metagenomics method from AGWs, whose complete genome was determined and genomic structure and phylogeny were fully characterized. The functional, biophysical and carcinogenic properties of HPV-JDFY01 need to be elucidated in the future research.

## Figures and Tables

**Figure 1 pathogens-11-01452-f001:**
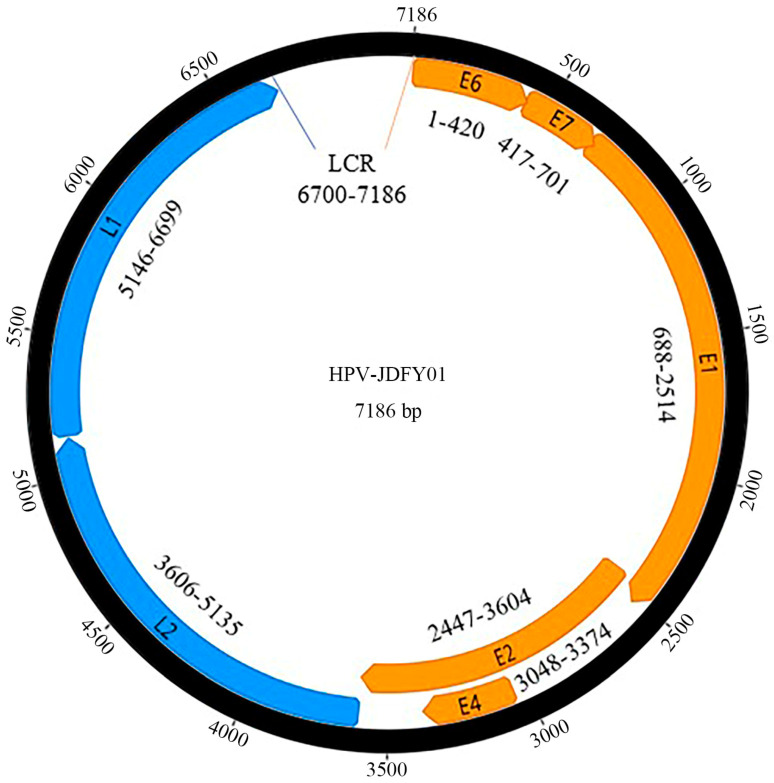
Genomic organization of HPV-JDFY01 isolated from AGW. Early proteins: E6 (nt 1–420, 420 bp); E7 (nt 417–701, 285 bp); E1 (nt 688–2514, 1827 bp); E2 (nt 2447–3604, 1158 bp) and E4 (nt 3048–3374, 327 bp). Late proteins: L2 (nt 3606–5135, 1530 bp) and L1 (nt 5146–6699, 1554 bp), and a 487 bp long control region (LCR) located at 6700 to 7186 of the genome between L1 and E6. The sizes of overlapping fragments are 4 bp between E6 and E7, 14 bp between E7 and E1, and 68 bp between E1 and E2. Even the entire E4 lies fully inside the E2 ORF.

**Figure 2 pathogens-11-01452-f002:**
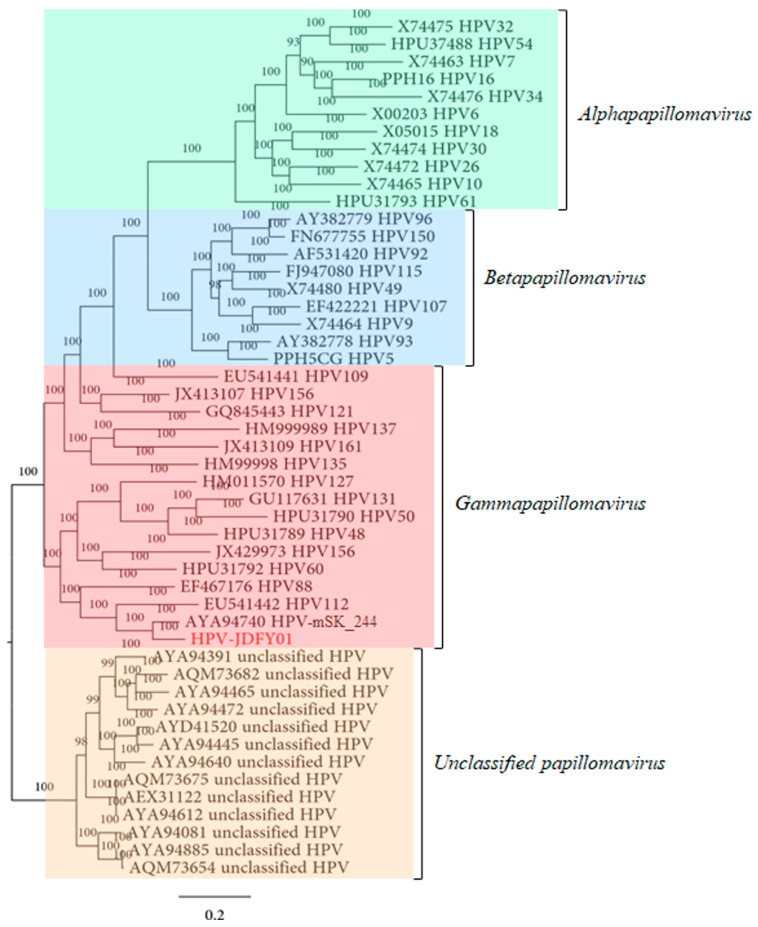
Phylogenetic analysis based on the amino acid sequence of HPV-JDFY01 L1 protein. HPV-JDFY01 is labeled in red.

**Figure 3 pathogens-11-01452-f003:**
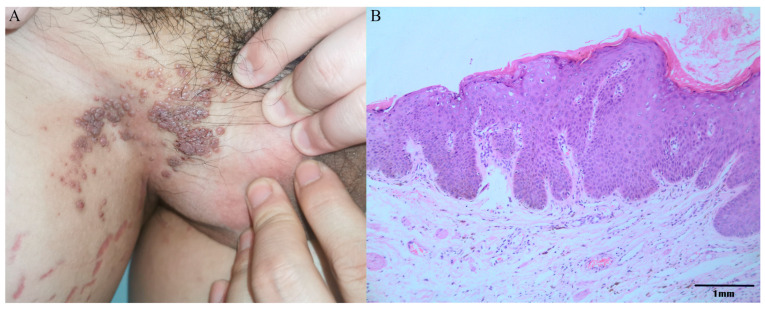
Clinical and pathological manifestations of AGW with HPV-JDFY01 positive by epidemiological screening. (**A**) Genital warts present as multiple, flat papules on the inguinal skin. (**B**) Representative pathological findings with HE staining in 100-fold magnification indicate hyperkeratosis, parakeratosis, acanthosis, papillomatosis, and characteristic vacuolated parakeratotic and granular cells (koilocytes).

**Table 1 pathogens-11-01452-t001:** Specific primer sequences used to screen and amplify the whole genome of HPV-JDFY01.

Primer ID	Application	Primer Sequences (5′-3′)
Screening WF	First round	AGGCCTCCATTACCTCTTCTG
Screening WR		AGCACAGATGGTTCAGTTGT
Screening NF	Second round	AGAGTTTGATCTGCGCCATTTGAT
Screening NR		AGATGATGGCGAGCTGG
Gap1 WF	First round	TTGCGCGCAAAAATGAGAGG
Gap1 WR		GAATTCCCCTGCTCCAGCTC
Gap1 NF	Second round	TTGCGCGCAAAAATGAGAGG
Gap1 NR		GCACACCGTGGACAAAGAAG
Gap2 WF	First round	AATCTGTCCCGTCTCCTGGA
Gap2 WR		TGGAACCCCGAATGTGGATG
Gap2 NF	Second round	AATCTGTCCCGTCTCCTGGA
Gap2 NR		TGCCAGCAGCTTTACAATGC
Gap3 WF	First round	TGGTCCACCAGATACCGGAA
Gap3 WR		TGGAAGTAGGAGTGGGCTGT
Gap3 NF	Second round	GCAAAACACAAACATCCACAGC
Gap3 NR		TGGAAGTAGGAGTGGGCTGT
Gap4 WF	First round	AGCTCAGCGCATGTAGTTTTG
Gap4 WR		ACCCGAAACTGAGAACCTGAC
Gap4 NF	Second round	CTCAGCGCATGTAGTTTTGCA
Gap4 NR		CCACTGGTTTTTGGGGAGGT
Gap5 WF	First round	TGGCAAACGAGAACAAGGTT
Gap5 WR		TCCTCCTCCTCTGCTTCAGT
Gap5 NF	Second round	TGCCAGACATTACTTTGCCAG
Gap5 NR		TGCAAGCTTTCATTCCGACA

WF: upstream primer of first round, WR: downstream primer of first round, NF: upstream primer of second round, NR: downstream primer of second round.

## Data Availability

The data presented in this study are available on request from the corresponding author.

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
