# Peer review of "Viral Metagenomics Reveals a Putative Novel HPV Type in Anogenital Wart Tissues"

_pathogens, 2022, doi:10.3390/pathogens11121452_

Round 1

Reviewer 1 Report

In this manuscript entitled “Viral metagenomics reveals a putative novel HPV type in anogenital wart tissues”, the authors conducted viral metagenomic and bioinformatic analysis and discovered a putative novel Gammapapillomavirus named HPV-JDFY01. The data are novel and informative. So, I recommend this manuscript for publication in Pathogens after responding to some minor comments, explained below.

Line 73,

“These sample pools were filtered through 0.45 μm filters (Millipore) to remove cellular particles in the sample.”

I can not understand the flow of the above experiment. If the samples were ground with beads, centrifuged, and the supernatants flowed through 0.45 μm filters, any intact infected cells would be caught. The authors used the flow-through liquid. What is the purpose of reacting the filter with DNase or RNase even though the sample trapped on the filter is not used?

I read References 10 and 11. They used 0.45 μm filters to collect only virus particles (virons) in the plasma pools to exclude anything larger than the virus. However, this study focused on HPV RNA, but HPV is a DNA virus, and the only thing packaged in capsid is viral DNA, not RNA; HPV RNA is expressed only in the host cell. In that sense, what is the significance of using a 0.45 μm filter in this study? If the authors wanted to collect only HPV particles, wouldn't the authors get viral RNA? Or did the authors want to collect only HPV RNA in the homogenized host cells? Please explain it.

Line 82,

Phres Score

Phred Score is correct.

Line133-135,

The authors mentioned the nucleotide position of each viral gene in these sentences. However, “bp” is not the correct expression. So please change the expression as follows. In addition, please add the nucleotide length of each viral gene.

Ex) E6 (nucleotide (nt) 1-420, 420 bp), E7 (nt 417-701, 285 bp)

Line 168-170 (the legend of Figure 1)

Please reword it about the nucleotide position of each viral gene in the same way described above.

line 175-184,

I have a question because I could not see the HPV subtype name in the phylogenetic trees in Figure 2.

Is there no difference in HPV-JDFY01 position in the phylogenetic tree among each gene? If there is any difference, please mention it in detail in each gene.

Figure 2,

The letters are too small to read the viral name.

I recommend that the authors separate it into each viral gene figure.

Ex, E6, Fig2; E7, Fig 3… or the authors had better choose only one gene in figure 2, and the others were added in the supplementary materials as supplementary figures.

Line 190-192,

Is the wart the same shape as the ordinary condyloma infected with HPV-6 or HPV-11? If there is a difference or characteristic, please note it.

Author Response

Dear Reviewer 1:

    On behalf of my co-authors, we are very grateful to you for giving us an opportunity to revise our manuscript. We appreciate you very much for your positive and constructive comments and suggestions on our manuscript entitled “Viral metagenomics reveals a putative novel HPV type in anogenital wart tissues” (Manuscript ID: pathogens-2029396).

We have studied reviewer’s comments carefully and tried our best to revise our manuscript according to the comments. In this revised version, changes to our manuscript were all highlighted within the document by using red colored text.

The following are the responses and revisions I have made in response to the reviewer’s questions and suggestions on an item-by-item basis. Thanks again to the hard work of the editor and reviewer!

Response to the comments of Reviewer 1

Point 1:

Line 73,

“These sample pools were filtered through 0.45 μm filters (Millipore) to remove cellular particles in the sample.”

I can not understand the flow of the above experiment. If the samples were ground with beads, centrifuged, and the supernatants flowed through 0.45 μm filters, any intact infected cells would be caught. The authors used the flow-through liquid. What is the purpose of reacting the filter with DNase or RNase even though the sample trapped on the filter is not used?

I read References 10 and 11. They used 0.45 μm filters to collect only virus particles (virons) in the plasma pools to exclude anything larger than the virus. However, this study focused on HPV RNA, but HPV is a DNA virus, and the only thing packaged in capsid is viral DNA, not RNA; HPV RNA is expressed only in the host cell. In that sense, what is the significance of using a 0.45 μm filter in this study? If the authors wanted to collect only HPV particles, wouldn't the authors get viral RNA? Or did the authors want to collect only HPV RNA in the homogenized host cells? Please explain it.

Response 1: Thank you for your comment. It is important to emphasize that the extraction method used (QIAamp Viral RNA Mini Kit (Qiagen)), which despite its name, extracts BOTH RNA and DNA. We have used this method numerous times to identify both RNA and DNA viruses as shown here. While a DNase step following extraction can be used to remove DNA we do not perform such digestion (nuclease is used only prior to extraction) and both RNA and DNA are extracted. We now clarify that this kit extracts both RNA and DNA. And we provide two more relevant references for easy understanding, as detailed in the Link.(https://pubmed.ncbi.nlm.nih.gov/27411548/ï¼›https://pubmed.ncbi.nlm.nih.gov/32234038/ ï¼‰

Point 2:

Line 82,

Phres Score

Phred Score is correct.

Response 2: Thanks for your careful checks. We feel really sorry for our carelessness. Based on your comments, we have corrected the “Phres Score” into “Phred Score”. (Page 2, Line 82)

Point 3:

Line133-135,

The authors mentioned the nucleotide position of each viral gene in these sentences. However, “bp” is not the correct expression. So please change the expression as follows. In addition, please add the nucleotide length of each viral gene.

Ex) E6 (nucleotide (nt) 1-420, 420 bp), E7 (nt 417-701, 285 bp)

Line 168-170 (the legend of Figure 1),

Please reword it about the nucleotide position of each viral gene in the same way described above.

Response 3: Thanks for your correction. We feel sorry for the incorrect expression. We have reworded all about the nucleotide position of each viral gene as you suggested. (Page 4, Line 135-138 and Page 5, Line 171-173)

Point 4:

Line 175-184,

I have a question because I could not see the HPV subtype name in the phylogenetic trees in Figure 2.

Is there no difference in HPV-JDFY01 position in the phylogenetic tree among each gene? If there is any difference, please mention it in detail in each gene.

Response 4: Thank you for your kind reminder. Our phylogenetic analysis was performed based on amino acid sequences. “AYA94740” is the protein ID of HPV-mSK_244 L1. “AYA94739” is the protein ID of HPV-mSK_244 L2. And so on for the rest. In each phylogenetic tree, HPV-JDFY01 shares the highest sequence identity with HPV-mSK_244 and is closest to it. Thus, there is no difference in HPV-JDFY01 position in the phylogenetic tree.

Point 5:

Figure 2,

The letters are too small to read the viral name.

I recommend that the authors separate it into each viral gene figure.

Ex, E6, Fig2; E7, Fig 3… or the authors had better choose only one gene in figure 2, and the others were added in the supplementary materials as supplementary figures.

Response 5: We are grateful for your suggestion. From the perspective of aesthetic typography, we unanimously decided to select the L1 gene in figure 2. (Page 6, Line 189-190) And the others have been added in the supplementary materials as supplementary figures. (Page 5, Line 182-183)

Point 6:

Line 190-192,

Is the wart the same shape as the ordinary condyloma infected with HPV-6 or HPV-11? If there is a difference or characteristic, please note it.

Response 6: Thank you for your question. But their clinical presentation was similar, with no significant differences.

Again, thank you for giving us the opportunity to strengthen our manuscript with your valuable comments and queries. We have worked hard to incorporate your feedback and hope that these revisions persuade you to accept our submission.

Sincerely,

Hui-min Hu MD.,

Department of Dermatology,

Affiliated Hospital of Jiangsu University,

Zhenjiang, Jiangsu 212001, China,

E-mail: huhuimin1124@126.com

Reviewer 2 Report

The study describes an investigation for new strains of HPV. The method, results, and discussion sections clearly present the objectives and outcomes of the study. The following are some suggestions to improve:

1. Present the other HPV types identified in the 110 wart samples.

2. Specify further details on how "low sequence similarity" (line 92, Methods) is defined and discovered. Add a sequence diagram if needed.

3. Modify the description of figure 1, and use medical terminologies and proper histopathologic description of the H&E-stain section.

Minor grammatical errors must also be corrected throughout the manuscript.

Author Response

Dear Reviewer 2:

    On behalf of my co-authors, we are very grateful to you for giving us an opportunity to revise our manuscript. We appreciate you very much for your positive and constructive comments and suggestions on our manuscript entitled “Viral metagenomics reveals a putative novel HPV type in anogenital wart tissues” (Manuscript ID: pathogens-2029396).

We have studied reviewer’s comments carefully and tried our best to revise our manuscript according to the comments. In this revised version, changes to our manuscript were all highlighted within the document by using red colored text.

The following are the responses and revisions I have made in response to the reviewer’s questions and suggestions on an item-by-item basis. Thanks again to the hard work of the editor and reviewer!

Response to the comments of Reviewer 2

Point 1: Present the other HPV types identified in the 110 wart samples.

Response 1: Thank you for your comment. The other HPV types, which were identified in the 110 wart samples, have been presented in Supplementary Table 1. (Page 4, Line 125-127)

Supplementary Table 1. The other HPV types identified in the 110 wart samples

Genus

Species

Serotype

Alphapapillomavirus

Alphapapillomavirus 1

HPV32

Alphapapillomavirus 3

HPV62, HPV81, HPV84

Alphapapillomavirus 4

HPV27b, HPV57b, HPV57c

Alphapapillomavirus 5

Alphapapillomavirus 6

HPV39, HPV53

Alphapapillomavirus 7

HPV45, HPV59, HPV68b

Alphapapillomavirus 8

HPV7, HPV40, HPV43, HPV91

Alphapapillomavirus 9

HPV31, HPV35, HPV52, HPV58, HPV67

Alphapapillomavirus 10

HPV6, HPV11, HPV44, HPV74

Alphapapillomavirus 14

HPV90

Gammapapillomavirus

Gammapapillomavirus 6

HPV108

Point 2: Specify further details on how "low sequence similarity" (line 92, Methods) is defined and discovered. Add a sequence diagram if needed.

Response 2: We are grateful for your suggestion. Six contigs from GW05 pool were aligned using BLASTx in the NCBI database, and we found that it had less than 60% homology with other known HPV sequences. Previous literatures considered that HPVs of the same species share ≥ 71% L1 similarity. Thus, it showed low sequence similarity to known HPVs.

Point 3: Modify the description of figure 1, and use medical terminologies and proper histopathologic description of the H&E-stain section.

Response 3: Thanks for your careful checks. We feel sorry for the incorrect expression. We have reworded about description of figure 1 and 3 as you suggested. (Page 5, Line 171-173 and Page 6, Line 197-200)

Point 4: Minor grammatical errors must also be corrected throughout the manuscript.

Response 4: Thank you for your kind reminder. We have checked the grammar of the full text.

Again, thank you for giving us the opportunity to strengthen our manuscript with your valuable comments and queries. We have worked hard to incorporate your feedback and hope that these revisions persuade you to accept our submission.

Sincerely,

Hui-min Hu MD.,

Department of Dermatology,

Affiliated Hospital of Jiangsu University,

Zhenjiang, Jiangsu 212001, China,

E-mail: huhuimin1124@126.com
